# THE GANFATHER: CONTROLLABLE GENERATION OF MALICIOUS ACTIVITY TO EXPOSE DETECTION WEAKNESSES AND IMPROVE DEFENCE SYSTEMS

## ABSTRACT

Machine learning methods to aid defence systems in detecting malicious activity typically rely on labelled data. In some domains, such labelled data is unavailable or incomplete. In practice this can lead to low detection rates and high false positive rates, which characterise for example anti-money laundering systems. In fact, it is estimated that 1.7–4 trillion euros are laundered annually and go undetected. We propose *The GANfather*, a method to generate samples with properties of malicious activity, without label requirements. To go around the need for labels, we propose to reward the generation of malicious samples by introducing an extra objective to the typical Generative Adversarial Networks (GANs) loss. Ultimately, our goal is to enhance the detection of illicit activity using the discriminator network as a novel and robust defence system. Optionally, we may encourage the generator to bypass pre-existing detection systems. This setup then reveals defensive weaknesses for the discriminator to correct. We evaluate our method in two real-world use cases, money laundering and recommendation systems. In the former, our method moves cumulative amounts close to 250 thousand dollars through a network of accounts without being detected by an existing system. In the latter, we recommend the target item to a broad user base with as few as 30 synthetic attackers. In both cases, we train a new defence system to capture the synthetic attacks.

## 1 INTRODUCTION

Many aspects of our society become increasingly dominated by digital systems, in turn providing new opportunities for illicit actors. For example, digital banking enables clients to open bank accounts more easily but also facilitates complex money laundering schemes. It is estimated that undetected money laundering activities worldwide accumulate to €1.7–4 trillion annually (Lannoo & Parlour, 2021), while operational costs related to anti-money laundering (AML) compliance tasks incurred by financial institutions accumulate to $37.1 billion (Ray, 2021). Another example are recommender systems, which are often embedded in digital services to deliver personalised experiences. However, recommender systems may suffer from injection attacks whenever malicious actors fabricate signals (e.g., clicks, ratings, or reviews) to influence recommendations. These attacks have detrimental effects on the user experience. For example, a one-star decrease in restaurant ratings can lead to a 5 to 9 percent decrease in revenue (Luca, 2016).

The detection of such malicious attacks is challenging in the following aspects. In many cases, these illicit activities are adversarial in nature, where an attacker and a defence system adapt to each other's behaviour over time. Additionally, labelled datasets are unavailable or incomplete in certain domains due to the absence of natural labels and the cost manual of feedback. For example, besides the large amount of undetected money laundering, the investigation of detected suspicious activity is often far from trivial, resulting in a feedback delay that can last months.

To address these issues, we propose *The GANfather*, a method to generate examples of illicit activity and train effective detection systems without any labelled examples. Starting from unlabelled data which we assume to be predominantly legitimate, the proposed method leverages a GAN-like

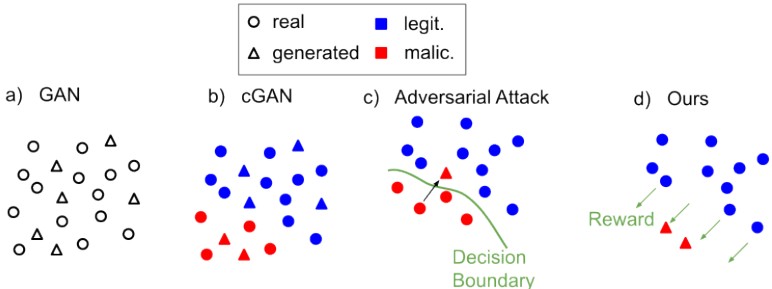

Figure 1: Comparing our method to some widely used approaches. **(a)** GAN: a vanilla GAN setup does not require any labels, but one cannot choose the class of a generated sample since the distribution of the data is learned as a whole. **(b)** conditional GAN (cGAN): using a cGAN, one learns the class-conditional distributions of the data, allowing the user to choose the class of a generated sample. However, labels are needed to train a cGAN. **(c)** Adversarial Attack (evasion): starting from a malicious example, perturbations are found such that a trained classifier is fooled and miss-classifies the perturbed example. While labels are typically required to select the initial example as well to train the classifier, eventually the adversarial attacks can be used to obtain a more robust classifier. **(d)** Ours: our method has some desirable properties from the three previous approaches: no labels are needed (as in a GAN), samples of a desired target class are generated (as in a cGAN) and a robust detection system can be trained (as in adversarial training). The combination of these properties in one framework is especially suitable in domains where no labelled data is available.

setup (Goodfellow et al., 2014) to train a generator which learns to create malicious activity, as well as a detection model learning to discriminate between real data and synthetic malicious data.

To be able to generate samples with malicious properties from legitimate data, our method includes an additional optimisation objective in the training loss of the generator. This objective is a use-case-specific, user-defined differentiable formulation of the goal of the malicious agents. Furthermore, our method optionally allows to incorporate an existing defence system as long as a differentiable formulation is possible. In that case, we penalise the generator when triggering existing detection mechanisms. Our method can then actively find liabilities in an existing system while simultaneously training a complementary detection system to protect against such attacks.

Our system makes the following assumptions and has the following desirable properties (in a context of adversarial attacks lacking labelled data):

**No labelled malicious samples are needed**. Here, we assume that our unlabelled data is predominantly of legitimate nature.
**Samples with features of malicious activity are generated**. The key to generate such samples from legitimate data is to introduce an extra objective function that nudges the generator to produce samples with the required properties. We implicitly assume that malicious activity shares many properties with legitimate behaviour. We justify this assumption since attackers often mimic legitimate activity to some degree, in order to avoid raising suspicious or triggering existing detection systems.
**A robust detection system is trained**. By training a discriminator to distinguish between the synthetic malicious samples and real data, we conjecture that the defence against a variety of real malicious attacks can be strengthened.

While each of these properties can be found separately in other methods, we believe that the combination of all the properties in a single method is novel and useful in the discussed scenarios. In Figure 1, we illustrate visually how our method distinguishes itself from some well-known approaches. Finally, while we only perform experiments on two use-cases (anti-money laundering and recommender systems) in the following sections, we believe that the suggested approach is applicable in other domains facing similar constraints, i.e., no labelled data and adversarial attacks, subject to domain-specific adaptations.

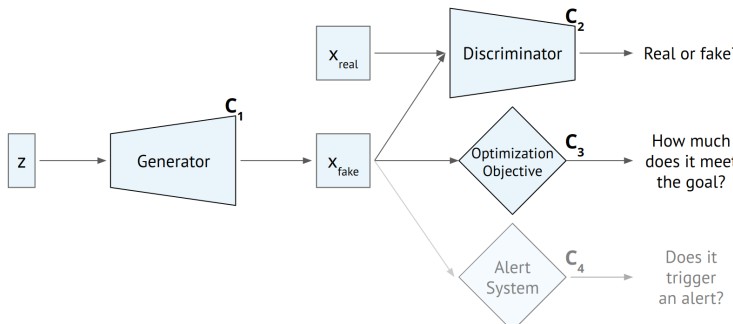

Figure 2: *The GANfather* framework. Its main components comprise a generator, $C_1$, which generates realistic attacks, a discriminator, $C_2$, which detects these attacks, an optimisation objective, $C_3$, to incentivise the generation of malicious instances. Finally, our method optionally supports the inclusion of an existing alert system, $C_4$.

## 2 METHODS

This section provides a general description of our proposed framework in Section 2.1. We proceed to describe two use-cases: anti-money laundering (AML) (Section 2.2) and detection of injection attacks in recommendation systems (Section 2.3).

### 2.1 GENERAL DESCRIPTION

Figure 2 depicts the general structure of our framework. It includes a generator, a discriminator, an optimisation objective, and, optionally, an existing alert system. Each component is discussed in more detail below.

**Generator.** As in the classical GAN architecture, the generator $G$ receives a random noise input vector and outputs an instance of data. However, unlike classical GANs, the loss of the generator $\mathcal{L}(G)$ is a linear combination of three components: the optimisation objective for malicious activity $\mathcal{L}_{Obj}(G)$, the GAN loss $\mathcal{L}_{GAN}(G, D)$ that additionally depends on the discriminator $D$, and the loss from an existing detection system $A$, $\mathcal{L}_{Alert}(G, A)$:

$$\mathcal{L}(G) = \alpha\mathcal{L}_{Obj}(G) + \beta\mathcal{L}_{GAN}(G, D) + \gamma\mathcal{L}_{Alert}(G, A) \tag{1}$$

where $\alpha$, $\beta$ and $\gamma$ are hyperparameters to tune the strength of each component. The last term is optional, and if no existing detection system is present we simply choose $\gamma = 0$. Note also that one of the parameters is redundant and we tune only two parameters in our experiments (or one if $\gamma = 0$).

We show theoretically, in a simplified setting, how this loss function changes the learning dynamics compared to a typical GAN in section A.3. Indeed, the stable point of convergence for the generator in our theoretical example moves away from the data distribution for any $\alpha > 0$.

**Discriminator.** The discriminator setup is the same as in a classical GAN. It receives an example and produces a score indicating the likelihood that the example is real or synthetic. Importantly, as explained in section A.3, the generator subject to equation 1 will generate data increasingly out of distribution for larger $\alpha$. Therefore, we do not require the discriminator accuracy to fall to chance level at training convergence, as is usual with GANs. Instead, the discriminator may converge to perfect classification and may be used as a detection system for illicit activity. In our experiments, we use the Wasserstein loss (Arjovsky et al., 2017) as our GAN loss.

**Malicious optimisation objective.** The optimisation objective quantifies how well the synthetic example is fulfilling the goal of a malicious agent. It can be a mathematical formulation or a differentiable model of the goal. This objective allows the generator to find previously unseen strategies to meet malicious goals.

**Alert system.** If an existing, differentiable alert system is present, we can add it to our framework to teach the generator to create examples that do not trigger detection (see equation 1). In that scenario, it is then beneficial for the discriminator to focus on the undetected illicit activity. Whenever the existing system is not differentiable, training a differentiable proxy may be possible.

**Generator vs. Discriminator views.** We note that, if required by the malicious optimisation objective, our generator can be adapted to generate samples which are only partially evaluated by the discriminator. For example, the layering stage of money laundering typically involves moving money through many financial institutions (FIs). However, detection systems are in place in single institutions and therefore have a reduced view of the entire operation. Our method can be adapted to capture this situation, by generating samples containing various fictitious FIs, but only sending the partial samples corresponding to each FI through the discriminator. Similarly, in recommender systems, the malicious objective can act on a group of synthetic illicit actors to generated coordinated attacks, while the detection of a fraudulent user is typically performed on a single-user level.

**Architecture optimisations.** In the next sections, we provide more details about the specific architectures used in the two experiments. We note that the architecture details (layer types, widths and number of layers) were first optimised using a vanilla GAN setup (i.e. setting $\alpha = 0, \beta = 1, \gamma = 0$ in equation 1). With the architecture fixed, the other hyperparameters were tuned as explained in the next sections.

## 2.2 ANTI-MONEY LAUNDERING (AML)

We tackle the layering stage of money laundering, in which criminals attempt to conceal the origin of the money by moving large amounts across financial institutions through what are known as "mule accounts". To represent the dynamic graph of transactions over time, we use a 3D tensor. The first two dimensions correspond to the accounts and the third dimension is discretised time. Each entry $A_{ijk}$ of the tensor corresponds to the cumulative amount sent from account $i$ to account $j$ on timestep $k$. For a more detailed overview of this representation, we refer the reader to section A.1.1.

**Architecture.** We implement the generator using a set of dense layers, followed by a set of transposed convolutions. Then, we create two branches: one to generate transaction amounts and the other to generate transaction probabilities. We use the probabilities to perform categorical sampling and generate sparse representations, similar to real transaction data. After the sampling step, the two branches are combined by element-wise multiplication, resulting in a final output tensor with the dimensions described above. More details of the generator's architecture are found in Table A1.

The discriminator receives two tensors with the same shape as inputs: one containing the total amount of money transferred per entry, and the other with the count information (mapping positive amounts to 1 and empty entries to 0). Each tensor passes through convolutional layers, followed by permutation-invariant operations over the internal and external accounts. Then, we concatenate both tensors. We reduce the dimensionality of the resulting vector to a classification outcome using dense layers. More details of the discriminator architecture are found in Table A2.

**Money Mule objective.** To characterise the money flow behaviour of layering, where money is moved in and out of accounts while leaving little behind, we define the objective function as the geometric mean of the total amount of incoming $(G(z)_{in})$ and outgoing $(G(z)_{out})$ money per generated account (Equation 2).

$$\mathcal{L}_{Obj}(G) = - \int \sqrt{G(z)_{in} \times G(z)_{out}} \cdot p(z) dz \tag{2}$$

Here $z$ stands for a random noise input to the generator $G$ and $p(z)$ stands for its probability distribution. This objective function incentivizes the generator to increase the amount of money sent and received per account and keep these two quantities similar.

**Existing Alert System.** In AML, it is common to have rule-based detection systems. In our case, the rules detection system contains five alert scenarios, capturing known suspicious patterns such as a sudden change in behaviour or rapid movements of funds. However, these rules are not differentiable, and our generator requires feedback in the form of a gradient. Hence, we construct a deep learning model as a proxy for the rules system. We hard-code a neural network mimicking the

rules' logic operations by choosing the weights, biases and activation functions appropriately. This network gives the same feedback as the rules system would, but in a differentiable way.

**Diversity measure.** To quantify the diversity of the generated attacks of a generator $G$, we calculate the inception score of three distributions extracted from the generated data: the amount distribution $d_a$; the count distribution $d_c$ (number of transactions per account); and the interval distribution $d_i$ (the time difference between consecutive transactions of a same account). We then define the diversity score $S_{div}$ as the average of these three inception scores (Equation 3, $D_{KL}$ denotes the KL divergence).

$$S_{div} = (\mathbb{E}_{x \sim p_G}[D_{KL}(d_a(x)||\mathbb{E}_{y \sim p_G}[d_a(y)])]+ \\ \mathbb{E}_{x \sim p_G}[D_{KL}(d_c(x)||\mathbb{E}_{y \sim p_G}[d_c(y)])]+ \\ \mathbb{E}_{x \sim p_G}[D_{KL}(d_i(x)||\mathbb{E}_{y \sim p_G}[d_i(y)])])/3 \tag{3}$$

## 2.3 RECOMMENDATION SYSTEM

In this work, we consider collaborative filtering recommender systems. However, our method is compatible with any other differentiable recommender. The system receives a matrix of ratings $R$ with shape $(N_u, N_i)$, where $N_u$ is the number of users and $N_i$ is the number of items. First, we compute cosine distances between users, resulting in the matrix $D$ of shape $(N_u, N_u)$. Then, we compute the predicted ratings $P$ as a matrix product between $D$ and $R$ [1]. We also note that, unlike in the AML scenario, we do not have an existing detection system in this setup.

**Architecture.** The generator consists of multi-layer perceptrons that gradually increase the size of the random noise input vector to the output vector of ratings. We implement the network using residual blocks composed of two dense layers within the skip connection. Then, similarly to the AML implementation, we create two branches: one for ratings and the other for probabilities. Each of these passes through additional residual blocks until the last dense layer, where we scale up the vector size to the number of items $N_i$, before performing the categorical sampling step. In our experiments, each synthetic user is independent, but the architecture could easily be adapted to generate a group of users from a single noise vector. More details of the generator architecture are available in Table A4.

The discriminator's architecture mirrors that of the generator. As in the AML implementation, it receives two tensors with the same shape: one containing the ratings and the other with the count information. We scale down each tensor with a dense layer before passing through residual blocks. Finally, we concatenate the two vectors into a single vector. After passing it through additional residual blocks, we scale down the final vector to single value output with a dense layer. Invariant permutation is not required because we generate and process each user separately. More details of the discriminator architecture available in Table A5.

**Injection Attack Objective.** We define the goal of malicious agents as increasing the frequency of recommendation of a specific item. The objective function in Equation 4 incentivizes the generator to increase the rating of the target item $t$ for every user.

$$\mathcal{L}_{Obj}(G) = \int \sum_i^{N_u} \sum_j^{N_i} (P_{ij}(z) - P_{it}(z))_+ \cdot p(z)dz \tag{4}$$

Here, the matrix of predicted ratings $P$ depends on the random inputs $z$ through the generator $G$ and $(\cdot)_+$ denotes a rectifier setting negative values to zero.

## 3 RESULTS

We evaluate the efficiency of *TheGANfather* to generate and detect attacks in two use-cases: money laundering (Section 3.1) and recommendation system (Section 3.2).

---

[1]We decide not to represent time since most classical recommender systems do not account for it. However, it is possible to temporal information using a similar setup to what we described in the AML use case.

### 3.1 MONEY LAUNDERING

**Setup.** We use a real-world dataset of financial transactions, containing approximately 200,000 transactions, between 100,000 unique accounts, over 10 months[2]. We implement *The GANfather*'s generator and discriminator following the architectures presented in section 2.2.

**Results.** We conduct a hyperparameter random search over the learning rate and the weights of the loss function components $\beta$ and $\gamma$ (see Equation 1) to study the impact of different the relative weights for each component, as shown in Table A3. It is important to note that we do not have a ground truth for the generated data, and therefore are limited to measure the success of the generators as how much money is flowing without being detected by the existing AML system, as well as how much variety we find in generated samples (details on the diversity score in equation 3). As shown in Figure 3, we did not find a trade-off between variety and money flowing, instead generators with high diversity score usually also circulated more money. In Figure 4, we compare the distribution of money flows from such a generator compared to the real data distribution. We can observe that the generated samples successfully move more money through the accounts than real data (up to 250k vs. up to 7k dollars respectively). Interestingly, the distribution of amounts used is similar to real data, and the main difference is the amount of transactions used (see Figure A2).

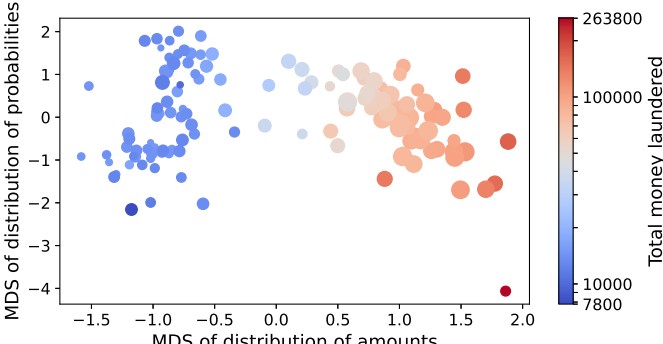

Figure 3: Hyperparameter search results. Each point corresponds to a different generator, for which generated samples did not trigger the existing detection system. Distances between generators are calculated on the empirical distributions of transaction amounts and transaction probabilities. The x and y axis denote the projected dimensions of a multidimensional scaling embedding of these distances. The size of the points relates to the diversity score of the data generated and the colour depicts the average total amount of money flowing per generated instance.

Next, we test the detection performance of the trained discriminators. To detect the potential bias that a discriminator has by being trained only on samples of the corresponding generator, we first build a *mixed* dataset where synthetic malicious data is sampled from various generators at various epochs during training and with different random noise seeds. We combine this synthetic dataset with real data, and use it to evaluate the trained discriminators. Importantly, no retraining on this mixed dataset is performed. In Figure 5, we observe that the discriminator typically achieves a perfect classification performance, especially for higher values of the $\beta$ parameter (see equation 1, note that $\alpha$ was fixed to a value of 1). This can be understood because the $\beta$ parameter limits the generated data distribution to diverge largely from the real data distribution. Therefore, discriminators trained with larger $\beta$ need to more accurately learn a decision boundary around the real data, in turn becoming more robust when evaluating on the mixed dataset.

### 3.2 RECOMMENDER SYSTEM

**Setup.** We use the MovieLens 1M dataset[3], comprised of a matrix of $6,040$ users and $3,706$ movies, with ratings ranging from 1 to 5 (Harper & Konstan, 2015). We implement the generator and discriminator and collaborative filtering recommender system as described in section 2.3. To compute

---

[2]Due to the confidential nature we cannot disclose the actual dataset

[3]https://www.kaggle.com/datasets/odedgolden/movielens-1m-dataset

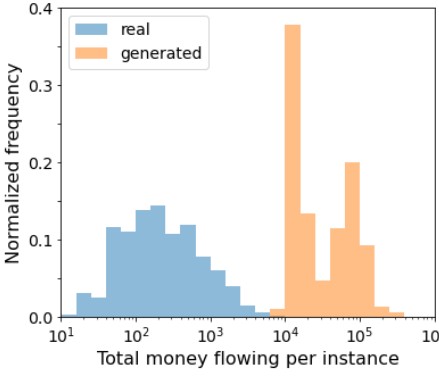

Figure 4: Empirical distribution of money flow (as defined by equation 2), for real data (blue) and generated data (orange).

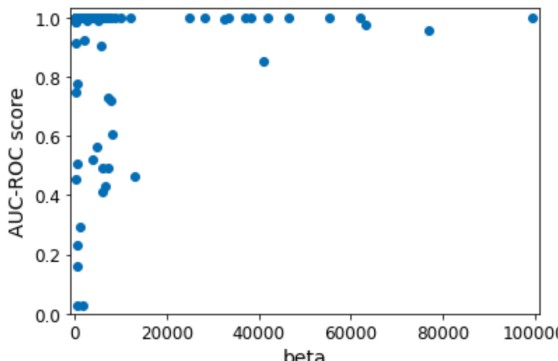

Figure 5: Discriminator AUC. Each point corresponds to a different discriminator.

the predicted ratings, during training we take a weighted average of ratings considering all users in the dataset. We consider all users during training because the initially generated ratings are random, and only providing feedback from the top-N closest users limits the strategies that the generator can learn. In contrast, we consider the top-400 closest neighbours to compute predicted ratings at inference since we observed empirically that this value produces the lowest recommendation loss.

In this scenario, we do not use an existing detection component, corresponding to $\gamma = 0$ in equation 1. We train our networks with 300 synthetic attackers but evaluate the generator's ability to influence the recommender system with injection attacks of various sizes. We also define four baseline attacks: (1) a rating of 5 for the target movie and 0 otherwise, (2) a rating of 5 for the target movie and $\sim$90 random ratings for randomly chosen movies, (3) a rating of 5 for the target movie and $\sim$90 random ratings for the top 10% highest rated movies, (4) a rating of 5 for the target movie and $\sim$90 random ratings for the top 10% most rated movies.

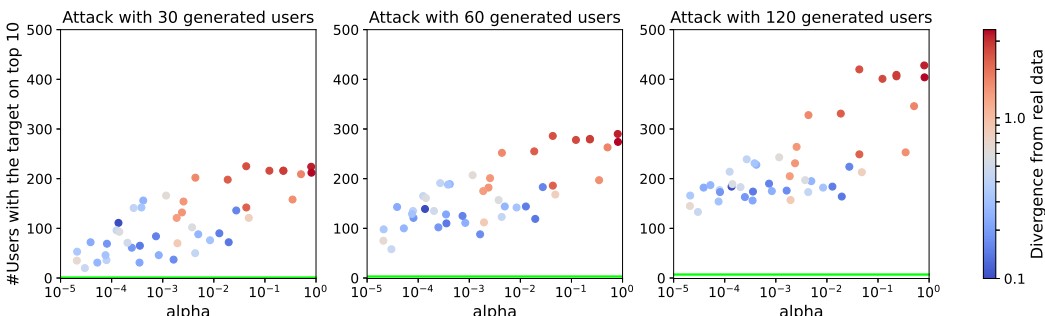

Figure 6: Hyperparameter search results. Each point corresponds to a different generator. The x-axis corresponds to $\alpha$, the y-axis to the number of users with the target item in their top-10 recommendations, and the colour denotes the KL divergence between real and generated rating profile distributions. We include the number of generated users above each panel. The green line represents the best baseline in each panel, which barely manages to increase the recommendation of the target movie to the other users.

**Results.** We choose $\beta = 1 - \alpha$ in equation 1, with $0 \leq \alpha \leq 1$ and perform a hyperparameter search over $\alpha$. In Figure 6, we observe that with as few as 30 synthetic attackers the generated attacks (dots) consistently outperform the best baseline (green line) in recommending the target movie. We observe that increasing $\alpha$ leads to generators whose attacks increasingly recommend the target movie, at the cost of moving further away from the real data distribution (measured through the KL divergence). Increasing the number of generated users also increases the target movie's recommendation frequency to real users.

Finally, we analyse the detection of synthetic attacks. As in the AML scenario we build a test set containing real and synthetic data, where the synthetic data contains a mixture of samples from various trained generators to identify the possible bias of a discriminator to attacks by the corresponding generator. We then quantify the AUC of the trained discriminators. In Figure 7, we observe that most discriminators achieve around 0.75 AUC. Unlike the AML scenario, this suggests that the discriminators are tuned to detect synthetic data from their respective generators, but less so from other generators. We tested whether we can train a better discriminator directly on this mixed test set, and indeed are able to obtain near-perfect classification (Figure 8).

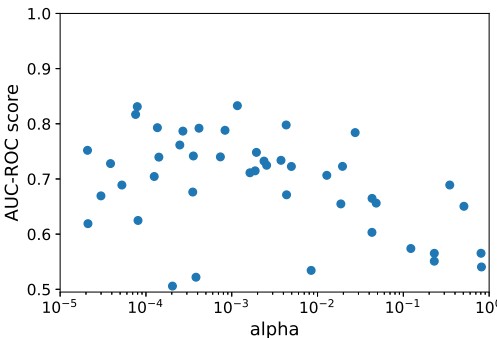 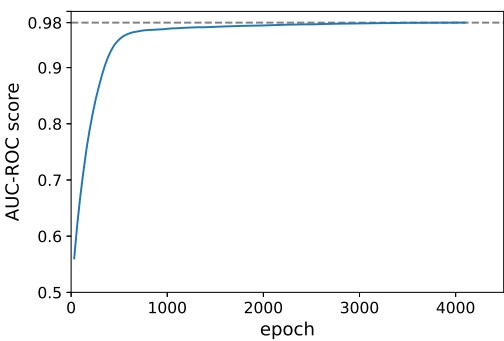

Figure 7: Discriminator AUC. Each point corresponds to a different discriminator which is evaluated on the test set without retraining.

Figure 8: We can train a new discriminator achieving almost perfect performance on the test set.

## 4 RELATED WORK

**Controllable data generation.** Wang et al. (2022) review controllable data generation with deep learning. Among the presented works, we highlight De Cao & Kipf (2018). It leverages a GAN trained with reinforcement learning to generate small molecular graphs with desired properties. Their work is similar to ours in that we both (1) extend a GAN with an extra objective and (2) use similar data representations, namely sparse tensors. However, whereas De Cao & Kipf (2018) uses a labelled dataset of molecules and their chemical properties, our method does not rely on any labelled data.

**Adversarial Attacks.** A vast amount of literature exists on the generation of adversarial attacks (see Xu et al. (2020) for a recent review). Such attacks have been studied in various domains and using various setups (e.g. cybersecurity evasion using reinforcement learning (Apruzzese et al., 2020), intrusion detection evasion using GANs (Usama et al., 2019), sentence sentiment misclassification using BERT (Garg & Ramakrishnan, 2020)). In all cases, a requirement is that labelled examples of malicious attacks exist.

**Anti-Money Laundering.** Typical anti-money laundering solutions are rule-based (Watkins et al., 2003; Savage et al., 2016; Weber et al., 2018). However, rules suffer from high false-positive rates, may fail to detect complex schemes, and are costly to maintain. Machine learning-based solutions tackle these problems (Chen et al., 2018). Given the lack of labelled data, most solutions employ unsupervised methods like clustering (Wang & Dong, 2009; Soltani et al., 2016), and anomaly detection (Gao, 2009; Camino et al., 2017). These assume that illicit behaviours are rare and distinguishable, which may not hold whenever money launderers mimic legitimate behaviour. Various supervised methods have been explored (Jullum et al., 2020; Raza & Haider, 2011; Lv et al., 2008; Tang & Yin, 2005; Oliveira et al., 2021), but most of these works use synthetic positive examples or incompletely labelled datasets. To avoid this, Lorenz et al. (2020) propose efficient label collection with active learning. Deng et al. (2009) and (Charitou et al., 2021) explore data augmentation using conditional GANs. Lastly, Li et al. (2020) and Sun et al. (2021) propose a metric to detect dense money flows in large transaction graphs, resulting in an anomaly score. Their method does not involve training of a classifier, and instead relies on generating many subsets of nodes and iteratively calculating the anomaly score.

**Recommender systems (RS) injection attacks.** Most injection attacks on RS are hand-crafted according to simple heuristics. Examples include random and average attacks (Lam & Riedl, 2004), bandwagon attacks (Burke et al., 2005a) and segmented attacks (Burke et al., 2005b). However, these strategies are less effective and easily detectable as most generated rating profiles differ significantly from real data and correlate with each other. Tang et al. (2020) address the optimisation problem of finding the generated profiles that maximise their goals directly through gradient descent and a surrogate RS. Some studies apply GANs to RS to generate attacks and defend the system. Wu et al. (2021) combines a graph neural network (GNN) with a GAN to generate their attack. The former select which items to rate, and the latter decides the ratings. Zhang et al. (2021) and Lin et al. (2022) propose a similar setup to ours in which they train a GAN to generate data and add a loss function to guide the generation of rating profiles. In both works, the main differences to our work are the usage of template rating profiles to achieve the desired sparsity, the chosen architecture and loss functions. In our work, sparsity is learned by the generator through the categorical sampling branch (see section 2). Moreover, our method allows the generation of coordinated group attacks by generating multiple attackers from a single noise vector.

## 5    CONCLUSION

In this work, we propose *The GANfather* to generate data of a novel class (malicious activity) without labelled examples, while simultaneously training a detection network to classify the novel class correctly. We performed experiments in two domains. In the anti-money laundering setting, the generated attacks are able to move up to 250,000 dollars using just five internal accounts, and without triggering an existing detection system. In the recommender system setting, we generate attacks that are substantially more successful at injection attacks than naive baselines. In both cases, we train a near-perfect classifier to detect the synthetic malicious activity. While no ground truth for the generated attacks are available, we argue that any attack strategy that is possible could in principle be exploited by attackers. While a real test in a deployment scenario is lacking and should be addressed in future work, we believe our current experiments provide a proof of value of the method. In these experiments, our method generates a variety of successful attacks, and we therefore believe it can be a valuable method to improve the robustness of defence systems.

The limitations of our method lie in its assumptions. Firstly, we assume that the unlabelled data is dominated by legitimate events, and our method would not work in settings where this is not the case. Secondly, we assume that we can quantify the malicious objective in terms of available features. In this case, one could argue we can just use the malicious objective as a detection score. However, the detection system often has a (much) smaller view than the malicious objective. For example, anti-money laundering systems only view incoming and outgoing transactions for *one* financial institution. However, our objective can be adapted to generate malicious activity mimicking flows across *multiple* synthetic financial institutions, while keeping the view of the discriminator on an individual institution level. Thirdly, while our method does not prevent generated data to be very different from real data, we argue that the strength of our method is in generating more subtle attacks that are not immediately distinguishable from real data. Finally, while we chose the malicious objectives to be as simple as possible in our proof of concept experiments, there is no restriction to make them more complex as long as they are differentiable.

To conclude, our method fits the adversarial game between criminals and security systems by simulating various meaningful attacks. If existing defences are in place, our method may learn to avoid them and, eventually, train a complementary model. We hope our work contributes to increase the robustness of detection methods of illicit activity.

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

# A APPENDIX

## A.1 EXTENDED METHODS

### A.1.1 REPRESENTING DYNAMIC GRAPHS AS TENSORS

To represent the dynamic graph of transactions, we can use a 3D tensor as depicted in Figure A1. We assume the nodes of the dynamic graph are accounts, and the edges are transactions. The first two dimensions correspond to the weighted adjacency matrix of the accounts and the third dimension is time. We discretise the events into time windows of fixed length and group events that belong to the same entry in the tensor by summing their amounts. Our representation covers any dynamic network with a 3D tensor whose size is fixed and pre-specified, which allows us to avoid using recurrent models. While this approach may limit the size of generated data, domain experts reported that up to 95% of the money-laundering investigations involve cases containing up to 5 accounts.

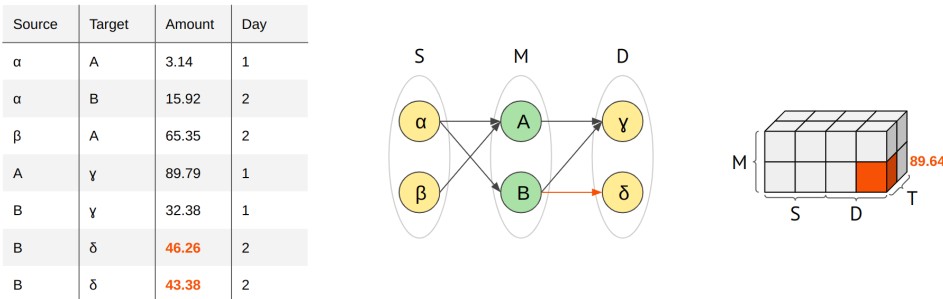

Figure A1: Data representation of transactional data. From the raw tabular data, we build the tripartite graph of the transactions, which is in turn represented as a 3D tensor.

### A.1.2 EXPERIMENTAL DETAILS: ANTI-MONEY LAUNDERING

| Index | Layer | Output shape |
|---|---|---|
| 0 | Linear(100, 400) | (400,) |
| 1 | Linear(400, 1600) | (1600,) |
| 2 | Linear(1600, 5000) | (5000,) |
|  | Reshape | (5,10,10,10) |
| 3 | ConvTranspose1d(10,10,4,2,1) | (5,10,20,10) |
| 4 | ConvTranspose1d(10,10,4,2,1) | (5,10,40,10) |
| 5 | ConvTranspose1d(10,10,4,2,1) | (5,10,80,10) |
| 6 | ConvTranspose1d(10,10,4,2,1) | (5,10,160,10) |
|  | Split amounts and probabilities | (5,10,160,10) |
| 7 | ConvTranspose1d(10,10,4,2,1) | (5,10,320,10) |
| 8 | Conv1d(10,1,1,1,0) | (5,10,320) |
|  | Categorical sampling | (5,10,320) |

Table A1: AML generator architecture.

| Index | Layer | Output shape |
|-------|-------|--------------|
| 0 | Conv1d(1,10,6,4,1) | (5,10,80,10) |
| 1 | Conv1d(10,10,6,4,1) | (5,10,20,10) |
| 2 | Conv1d(10,5,6,4,1) | (5,10,5,5) |
| | Reshape | (5,2,5,25) |
| | Mean pooling | (1,2,1,25) |
| | Concatenate amounts and probabilities | (100,) |
| 3 | Linear(100,128) | (128,) |
| 4 | Linear(128,32) | (32,) |
| 5 | Linear(32,1) | (1,) |

Table A2: AML discriminator architecture.

| $\alpha$ | 1 |
|----------|---|
| $\beta$ | $[10^2, 10^5]$ |
| $\gamma$ | $[10^3, 4 \times 10^3]$ |
| learning rate | $[10^{-4}, 3 \times 10^{-3}]$ |

Table A3: Hyperparameter search space on the AML use case.

### A.1.3 EXPERIMENTAL DETAILS: RECOMMENDER SYSTEMS

| Index | Layer | Output shape |
|-------|-------|--------------|
| 0 | ResBlock(128,128) | (128,) |
| 1 | ResBlock(128,128) | (128,) |
| | Split ratings and probabilities | (128,) |
| 2 | ResBlock(128,128) | (128,) |
| 3 | ResBlock(128,64) | (64,) |
| 4 | Linear(64,3706) | (3706,) |
| | Categorical sampling | (3706,) |

Table A4: RS generator architecture.

| Index | Layer | Output shape |
|-------|-------|--------------|
| 0 | Dense(3706,64) | (64,) |
| 1 | ResBlock(64,128) | (128,) |
| 2 | ResBlock(128,64) | (64,) |
| | Concatenate ratings and probabilities | (128,) |
| 3 | ResBlock(128,128) | (128,) |
| 4 | ResBlock(128,128) | (128,) |
| 5 | Linear(128,1) | (1,) |

Table A5: RS discriminator architecture.

| $\alpha$ | $[10^{-5}, 1]$ |
|----------|----------------|
| learning rate | $[10^{-5}, 3 \times 10^{-4}]$ |

Table A6: Hyperparameter search space on the RS use case.

## A.2 EXTENDED RESULTS

In Figure A2, we compare the distributions of the total amount of money flowing through the internal accounts from real data with our generated dataset; we observe that the generators consistently move more money than real accounts. By investigating the distributions of amounts per transaction and

the number of transactions per account, we verify that the generators can circulate more money than real accounts because they make more transactions but keep the amounts in a similar range as the real data.

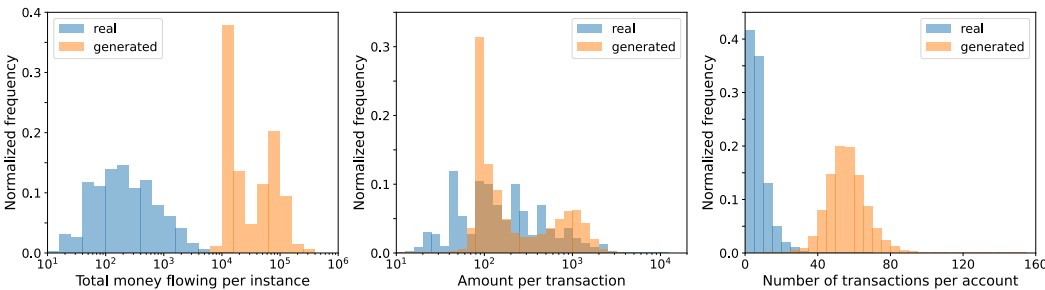

Figure A2: Comparison distributions of total money flow, amounts and counts between G and real.

### A.3 THEORETICAL JUSTIFICATION: SIMPLIFIED SETTING

In this section, we provide a simplified example to discuss certain aspects of our setup. We will assume no existing detection system is available ($\gamma = 0$ in equation 1). In the case such a system would be available, we assume its effect is to limit how far the generated data distribution can be from the real data distribution. Furthermore, we assume that a malicious objective would promote a change in the distribution of at least one feature of the generated data compared to the real data.

In order to facilitate the analytical calculations, we make the following simplifying assumptions. Firstly, we assume that our data consists of only one feature, for which the regular (legitimate) activity is distributed following a normal distribution $p_{\text{data}}$ with mean $\mu_d$ and standard deviation $\sigma_d$:

$$p_{\text{data}} = \mathcal{N}\left(\mu_d, \sigma_d\right) \tag{5}$$

Secondly, we assume that we do not have any samples of malicious activity but that we know that it is characterised by larger values of this feature compared to the legitimate activity. Thirdly, we assume that the generated data follows a normal distribution $p_{\text{gen}}$ with mean $\mu_g$ and standard deviation $\sigma_g$. Using $\gamma = 0$ and $\beta = 1 - \alpha$ in equation 1, assuming $0 \leq \alpha \leq 1$, we can write the training criterion of the generator as:

$$\mathcal{L}(G) = (1 - \alpha) \cdot (2 \cdot \text{JSD}\left(p_{\text{data}}|p_{\text{gen}}\right) - \log(4)) - \alpha\mu_g \tag{6}$$

Where the first term denotes the GAN loss Goodfellow et al. (2014) and the second term denotes our *malicious objective* rewarding the generator to produce samples with properties of the malicious data.

We can analytically solve the Jenson-Shannon Divergence (JSD) between the normal distributions, using $\sigma_m^2 = \sigma_d^2 + \sigma_g^2$,

$$\text{JSD}\left(p_{\text{data}}|p_{\text{gen}}\right) = \frac{1}{2}\text{KL}\left(p_{\text{data}}|0.5 * (p_{\text{data}} + p_{\text{gen}})\right) + \frac{1}{2}\text{KL}\left(p_{\text{gen}}|0.5 * (p_{\text{data}} + p_{\text{gen}})\right) \tag{7}$$

$$= \frac{1}{2}\left[\log\frac{\sigma_m}{\sigma_d} + \frac{\sigma_d^2 + (\mu_d - 0.5(\mu_d + \mu_g))^2}{2\sigma_m^2} - \frac{1}{2}\right. \tag{8}$$

$$\left. + \log\frac{\sigma_m}{\sigma_g} + \frac{\sigma_g^2 + (\mu_g - 0.5(\mu_d + \mu_g))^2}{2\sigma_m^2} - \frac{1}{2}\right] \tag{9}$$

From this, we can calculate the gradient w.r.t. $\mu_g$:

$$\frac{\partial \text{JSD}(p_{\text{data}}|p_{\text{gen}})}{\partial \mu_g} = \partial \left( \frac{1}{2} \left[ \log \frac{\sigma_m}{\sigma_d} + \frac{\sigma_d^2 + (\mu_d - 0.5(\mu_d + \mu_g))^2}{2\sigma_m^2} - \frac{1}{2} \right. \right. \tag{10}$$

$$\left. \left. + \log \frac{\sigma_m}{\sigma_g} + \frac{\sigma_g^2 + (\mu_g - 0.5(\mu_d + \mu_g))^2}{2\sigma_m^2} - \frac{1}{2} \right] \right) / \partial \mu_g \tag{11}$$

$$= \frac{1}{2} \partial \left( \frac{(0.5\mu_d - 0.5\mu_g)^2}{2\sigma_m^2} + \frac{(0.5\mu_g - 0.5\mu_d)^2}{2\sigma_m^2} \right) / \partial \mu_g \tag{12}$$

$$= \frac{\mu_g - \mu_d}{4\sigma_g^2 + 4\sigma_d^2} \tag{13}$$

Combining equations 6 and 13, we find that the gradient of the training objective of the generator w.r.t. the mean of the generated distribution $\mu_g$ is

$$\frac{\partial \mathcal{L}(G)}{\partial \mu_g} = \frac{(1 - \alpha)}{2} \frac{\mu_g - \mu_d}{\sigma_g^2 + \sigma_d^2} - \alpha \tag{14}$$

Without loss of generality, we set $\sigma_g^2 + \sigma_{\text{data}}^2 = k/2$, such that

$$\frac{\partial \mathcal{L}(G)}{\partial \mu_g} = (1 - \alpha)\frac{\mu_g - \mu_d}{k} - \alpha \tag{15}$$

Denoting as $\frac{\partial \mu_g}{\partial t}$ the changes of $\mu_g$ over time (i.e. a continuous version of the discrete gradient updates), and with $\eta$ a learning rate, this leads to the following linear dynamical system which we can analyse in function of $\mu_g$, $\mu_{\text{data}}$ and the hyperparameter $\alpha$:

$$\frac{\partial \mu_g}{\partial t} = -\eta \frac{\partial \mathcal{L}(G)}{\partial \mu_g} \tag{16}$$

$$\frac{\partial \mu_g}{\partial t} = -\eta(1 - \alpha)\frac{\mu_g - \mu_d}{k} + \eta\alpha \tag{17}$$

$$\frac{\partial \mu_g}{\partial t} = -\eta d\mu_g + \eta d\mu_d + \eta\alpha \tag{18}$$

Where we defined $d = (1 - \alpha)/k$. The stability of this linear system is defined by the sign of $-d$, which is always negative and hence the system has a stable fixed point. The stable fixed point for this dynamical system is easily found to be

$$\mu_g^\star = \mu_d + \frac{\alpha}{1 - \alpha}k \tag{19}$$

We plot the phase diagram of the dynamical system in Figure A3, showing the fixed point in function of the parameter $\alpha$.

From this calculation on a simplified setting, we can conclude the following:

- For $\alpha > 0$, our generated data will move away from the real data distribution and increasingly comply with the malicious objective.
- Various values of $\alpha$ will generate various levels of deviation from the real data. When no ground truth is available to test the resulting system, hyperparameter tuning and empirical testing should be performed. Another limiting factor for the deviation from real data can be the inclusion of an already available detection system.
- When generated data deviates from real data, the discriminator will increasingly achieve a perfect performance even at training completion. This is a major difference to standard GAN training.

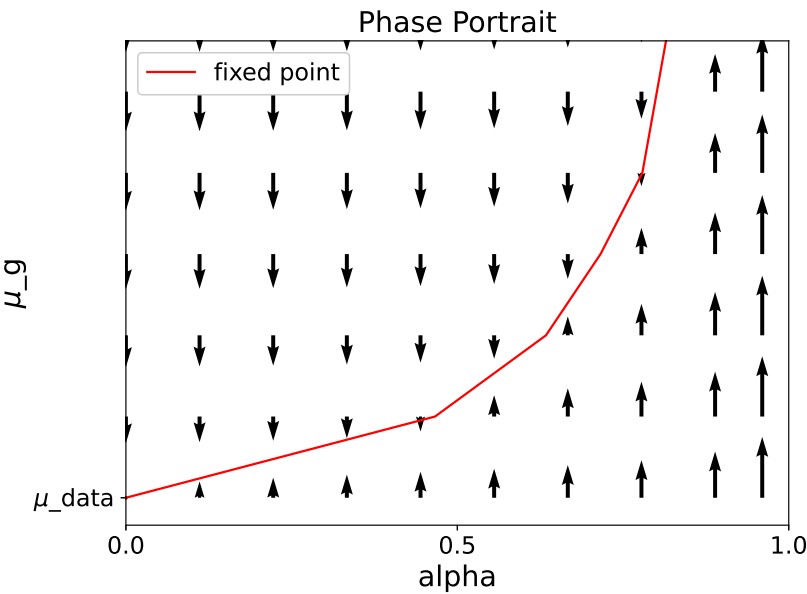

Figure A3: Phase portrait of our toy system. The fixed point of $\mu_g$ depends on hyperparameter $\alpha$. For $\alpha \to 1$, the fixed point approaches infinity. For $\alpha \to 0$, the fixed point converges to $\mu_d$. Arrows denote the direction of the gradient $\frac{\partial \mu_g}{\partial t}$

