# OpenReview forum: "The GANfather: Controllable generation of malicious activity to expose detection weaknesses and improve defence systems."
_ICLR.cc/2023/Conference — Submitted to ICLR 2023_

### Official Review · Reviewer_hdrh · 2022-10-18

**Confidence:** 5
**Correctness:** 1
**Technical Novelty And Significance:** 1
**Empirical Novelty And Significance:** Not applicable
**Recommendation:** 6

**Clarity, Quality, Novelty And Reproducibility:**

Overall, the presentation of the paper is inappropriate for ICLR.

The quality of the English text is mediocre: the manuscript is readable, but the writing is superficial at best.

Figures and Tables are of poor quality.

The topic addressed by the manuscript is source of abundant literature but appropriate for ICLR.

The references are not appropriate, and there are many related works missing.

The contribution is not significant.

The experiments are partially reproducible: the data are public, but the code is not publicly released.


**Strength And Weaknesses:**

STRENGTHS:
+ Potentially of use to practitioners


WEAKNESSES
- Misleading contributions
- Unfair claims
- Poor evaluation
- Lack of theoretical foundations
- No novelty
- Lack of comparison with prior work


**Summary Of The Paper:**

The paper tackles the problem of detection of “illicit” activities. The main proposal is “The Ganfather”, a Generative Adversarial Network (GAN) that automatically crafts samples conforming to a given “illicit activity” which – if not detected – will cause harm to the owners of an information system. By training a detector on the generated “illicit activities”, it is possible to develop a detector that protects the information system against such malicious samples. The proposal, for which no theoretical analysis is provided, is empirically evaluated on two datasets representing typical scenarios affected by illicit activities: money laundering, and recommender systems.

**Summary Of The Review:**

The paper cannot be accepted to ICLR, and I am confident of my decision. Truly, I believe the paper requires a complete re-write, starting from its conceptualization: put simply, the contributions of "The GanFather" - as presented in the paper - are unfair and misleading. Probably, the authors should simply begin by performing a detailed review of existing literature on “automatic” evasion of ML systems / generation of adversarial (or “out of distribution”) samples, as there are hundreds of papers that propose similar systems. Below I will list some additional issues that affect the paper.

**Misleading contributions (no labels).**
It is unfair to claim that the proposed method requires “no label” as a contribution. First, because the “Ganfather” is simply a GAN, which—by definition—requires no labels. Second, because the problem of labels is that they must be created via the human supervision: although, yes, the GanFather does not require supervision in the form of “labels”, it still requires supervision to define the “intended behavior [of the illicit activity]” (quoted from the introduction). Hence, the absence of labels is a "fake" contribution.

**Misleading contribution (automated attack generation).**
Proposing methods that “automatically generate attacks” is (1) not novel and (2) not a contribution per-se. There are literally thousands of approaches that aim to automatically generate attack samples, which may leverage either basic classifiers [C], or GANs [B], or Reinforcement Learning [A].

**Misleading contribution (detection).**
It is unfair to claim that the proposed method can “automatically learn from attacks” as a contribution. Indeed, the process of “re-training” on evasive samples is well-known in literature [A]. This also applies for the contribution denoted as “Expose and augment current defence system:” (which is a 1:1 overlap with the second one)

**Misleading contributions (generality).**
It is unappropriate to claim “generality” when the method is provided “as is” (there is a significant lack of theoretical analyses supporting its scientific soundness) and when such supposed “generality” is empirically evaluated on just two settings. What is worse, is that such experiments are shallow: I would have at least expected some sort of statistical validation, but there is no trace of such testings.

**The contributions are not contributions.**
Irrespective of the above, the "contributions" listed in the Introduction are not actually contributions, but rather properties of the proposed "The GANFather" method. The question is: does "The GANFather" perform better than existing methods? Unfortunately, the answer to this question cannot be answered because there is no hard-comparison with any "automatic generation" method. A proper comparison could be made by using any "GAN"-based method to generate some "illicit activities", and then showing that such activities would be either easily detected; or would not be good enough to make a detector "as robust" as those generated by "The GANFather".

**Lack of theoretical analysis.**
As a direct consequence of the above issue, there is no theoretical argument supporting not only the “generality”, but also the rationale behind the proposed “Ganfather” method. The description is provided in Section 2.1.1, but it only consists in a single equation (with several terms never introduced), with no justification whatsoever as to why it would even remotely work. What is worse is that the paper attempts to propose such a method as “novel”: as a matter of fact, no work is cited in Section 2.1 (aside from (Arjovsky et al. 2017) to regulate the GAN loss). I will be blunt and state the following: the claimed generality is passable as a “we use GANs to solve this problem, and hence our method is general; however, the specific application of our method requires domain expertise and significant tuning and supervision” which is unfair, unscientific and definitely not appropriate for ICLR.

**Misleading evaluation of “real-world”**
The abstract states that the paper “evaluates the method in two real-world use cases”. I was hoping that the evaluation was truly performed on some “real world” deployment. However, these use-cases simply use “benchmark” datasets collected from the real-world – which is hardly passable as an “evaluation on a real-world use case”. With such premises, any evaluation can be claimed to be a “real-world use case”.

Some additional issues:

•	The first paragraph of the Introduction is illogical: “Illicit activities frequently target digital systems and services. Importantly, these illicit activities are adversarial: an attacker and a defence system constantly adapt to each other’s behaviour.” First, “illicit activities” do not “target” anything: it is attacker who “target” digital systems and perform “illicit activities” with such systems. Second, the term “adversarial” is misleading, and it is not true that “attacker and defence system constantly adapt” (I dare say in most domains there is no adaptation by the “defence” at all). I suggest revising this paragraph entirely.

•	The second paragraph of the Itntroduction does not logically follow the previous one. What is the need of “for instance”? To me, this paragraph should follow something along the lines of “Despite providing many advantages, information systems are vulnerable to cyber attacks.” The same applies for the third paragraph of the Introduction.

•	The paper states that “Recent estimates indicate undetected money laundering activities of C0.7–3 trillion annually (Lannoo & Parlour, 2021)”. I checked (Lannoo & Parlour, 2021) and I couldn’t find a single occurrence of “0.7—3 trillion”. Rather, I found “Money laundering is estimated to cost the global economy between USD800 billion and USD2 trillion annually, according to the United Nations Office on Drugs and Crime report 2020”.

•	The following sentence in the Introduction should be revised “For example, a one-star decrease in restaurant ratings can lead to a 5 to 9 percent decrease in revenue (Luca, 2016)1. Detecting malicious agents is far from trivial. A critical challenge relates to class imbalance, as illicit activity is rare. Additionally, labelled datasets are often unavailable or incomplete due to the absence of natural labels and the cost of feedback, primarily generated through manual labelling.” The transition between the example and the technical problems is too abrupt.

•	The paper really looks like a “draft”. The term acronym “AML” is first mentioned in Section 2, but never introduced before (I originally believed it stood for “Adversarial Machine Learning”, but later found out it referred to “Anti-Money Laundering”). Parentheses are sometimes opened and not closed (e.g., beginning of Section 2.1). Equation 1 has several terms that are never mentioned, such as $G, O, D, A$.  The same applies for equation 2: what is $\mathcal{S}_{in}$?

•	What is the goal of generating “out of distribution” samples if such samples are not realistic in the first place? I am referring to the statement in Section 4.1: “Whereas De Cao & Kipf (2018) concerns generating realistic data that verifies some conditions (e.g., as our method could achieve leveraging the optional alert system), The GANfather generates out-of-distribution data to tackle adversarial domains)”



EXTERNAL REFERENCES

[A]: Apruzzese, G., Andreolini, M., Marchetti, M., Venturi, A., & Colajanni, M. (2020). Deep reinforcement adversarial learning against botnet evasion attacks. IEEE Transactions on Network and Service Management, 17(4), 1975-1987.

[B]: Usama, M., Asim, M., Latif, S., & Qadir, J. (2019, June). Generative adversarial networks for launching and thwarting adversarial attacks on network intrusion detection systems. In 2019 15th international wireless communications & mobile computing conference (IWCMC) (pp. 78-83). IEEE.

[C]: Garg, S., & Ramakrishnan, G. (2020, November). BAE: BERT-based Adversarial Examples for Text Classification. In Proceedings of the 2020 Conference on Empirical Methods in Natural Language Processing (EMNLP) (pp. 6174-6181).

========

Update after authors' response:

I appreciate the authors' efforts in improving the paper and toning down some of its original "overclaims". The paper now is more truthful as to what it does. I will increase my score substantially: from a 1 to a 6.

---

> ### Author Response · Authors · 2022-11-09
> **Clarifications on novelty and claims**
>
> We thank the reviewer for the detailed feedback. We regret the above judgment, but we believe that the reviewer missed some important information. Mainly, we believe that the reviewer erroneously interpreted the purpose of our work to be a typical ‘adversarial attack’ setting where the goal is to find the smallest possible perturbations to known samples that lie on the other side of the decision boundary of a trained classifier. In this setup, (1) labels are required to train the classifier and choose the samples to modify (2) there is no guarantee that the generated samples contain some properties of malicious activity. In fact, the goal of these attacks is typically to fool the classifier in producing a different label than a human would give.
>
> We avoid the label requirement by instead stimulating the generator to produce OOD examples that optimize a new objective. In this setup, we both guarantee that no labels are needed and that the generated samples have properties of malicious activity, which are two key differences from current approaches.
>
> **Misleading contributions (no labels).**
>
> We thank the reviewer for their remark. However, we disagree completely with the statement that this contribution is misleading. First of all, we are concerned with the generation of samples conditioned on some class label. While a classical GAN does not require labels to learn the data distribution, our situation is more similar to a conditional GAN which clearly requires labels. Moreover, we propose a method to go around the label requirement altogether, and still generate samples of a specified class (‘illicit activity’). The problem of defining the intended behavior indeed requires supervision. However, this requirement is clearly of a different scale than labeling thousands or millions of potentially suspicious activities. In the case of Money Laundering, investigating a single suspicious case can take up several days up to months. Labeling a full dataset then quickly becomes infeasible or extremely time consuming and expensive. Instead, defining the intended behavior of the malicious actors in the form of a reward function, as we propose, is a way of side-stepping this problem and creating a semi-synthetic dataset (i.e. only the positive labels are synthetic). For this purpose, we indeed use a GAN architecture but crucially with some additional components. Due to this setup, we can train a defense system in domains where labeled datasets are basically non-existent, such as the anti-money laundering domain. In fact, it is a huge problem to incorporate supervised models in this domain exactly due to the lack of labels and the large label delays. We proposed the method in this context, and the fact that we do not require labels is therefore a key property and important contribution of the approach.
> In a simple (but unrealistic) analogy, a typical adversarial attack setting would for example search for the smallest perturbations to an image of a cat such that a trained classifier would label it as a dog. This perturbation would not be visible to the human eye, and exploits the fact that deep neural nets are very expressive and fit the decision boundary around a finite training data set. Instead, our goal is very different: following the previous analogy, in our case we would have only data of cats, as well as define some differences between cats and dogs in an objective function (e.g. larger size, different head shape, different fur colour etc). Then, only having the data of cats and the objective function, our setup would generate examples of dogs. Clearly, our approach does not require any labels and has a different objective than the typical ‘adversarial attack’ setting.
>
> We believe that these important points have been lost to the reviewer, and would welcome the reviewer to read our more detailed replies below and revision their feedback.

---

> > ### Author Response · Authors · 2022-11-09
> > **Continued clarifications**
> >
> > **Misleading contribution (automated attack generation)**
> >
> > We thank the reviewer for the remark and provided references. We were not aware of some references mentioned, which we will gladly include in future iterations of our work. While the similarities are undeniable, we’d like to point out some crucial differences between the previous works and ours:
> >
> > Reference [A]. The authors propose a reinforcement learning framework to generate synthetic attacks that evade a certain cybersecurity system. While we share the same goal, the authors in reference [A] define an action space of perturbations that an agent can introduce on a known malicious sample. The agent then continues to modify the sample until it is able to evade detection or after a maximum amount of failures. Our approach is different since our reward function is assumed to define the intended behavior of the malicious agents. In this way, no known malicious samples are needed at all.
> >
> > Reference [B]. The authors propose a GAN-based framework to generate synthetic attacks that evade an intrusion detection system. While the similarities between their approach and ours are obvious, the authors again need examples from both legitimate and counterfeited examples. In fact, while the Discriminator is trained using both classes, the generator is trained only on the malicious data. This again differs from our method, where we do not require any malicious samples and instead define an objective for the malicious activity.
> >
> > Reference [C]. The authors propose a BERT-based method for generating adversarial examples in text. We believe that this method is relatively unrelated to our objectives. The goal of the authors is to perturb text such that (1) the modified text is indiscernible from the original by humans (2) classifier (e.g. sentiment analysis) performance is drastically reduced. The objective is therefore to find perturbations that are (1) very likely and (2) with the same intent as the original data. Our intent is to find perturbations which (1) are unlikely events in actual data and (2) change the intent of the original data (from legitimate activity to illicit) while obeying some objective function characterizing malicious activity.
> >
> > In summary, while the literature on the topic of adversarial attacks is vast, as the reviewer pointed out, we believe our method has some key differences to previous work. Please see our previous answer for more in-depth discussion.
> >
> > **Misleading contribution (detection)**
> >
> > We agree with the reviewer that the individual contribution claims by themselves may not be novel, but as argued in our previous responses, the pipeline as a whole (i.e. the combinations of all contributions in one method) are novel as far as we know. We have only encountered two works with similar properties, tailored to recommendation systems (i.e. Zhang et al. (2021) and Lin et al. (2022) cited in the related work section).
> >
> > **Misleading contributions (generality)**
> >
> > We thank the reviewer for the valuable feedback. We believe that our work provides a recipe that can be useful in the specific settings discussed above (no labels for the malicious class). In that sense, we believe that this recipe of using a GAN with the added components (objective characterizing malicious activity and optionally feedback from an existing detection system) can be applied to domains which face similar constraints. In that sense we meant “generality”. We do agree that there is no theoretical analysis of the proposed system and that the experiments are limited to the mentioned datasets. We will clarify and rephrase this claim in future revisions, as well as try to add more theoretical justification of our GAN system with the extra loss.

---

> > > ### Author Response · Authors · 2022-11-09
> > > **Continued clarifications**
> > >
> > > **The contributions are not contributions**
> > >
> > > We thank the reviewer for the valuable feedback. Comparing our method with existing methods is unfortunately hard, since to our knowledge we are the first to propose a method that does not require labels of the malicious activity at least in the anti-money laundering domain. We have found two works in the recommendation literature (and mentioned in our Related Work section) that propose similar ideas, namely Zhang et al. (2021) and Lin et al. (2022). It would be interesting to compare our methods against their systems on the Recommender System dataset.
> > >
> > > Furthermore, and as discussed in our replies above, the typical GAN adversarial attack setting is different from ours. The former does not guarantee that the generated examples contain properties of malicious activity, in fact the objective typically is to find the smallest perturbation that crosses the decision boundary of that specific classifier. This is quite different from our objective, where we (1) do not require labeled data (i.e. an existing classifier) (2) generate examples that contain properties of malicious activity due to the additional objective function. Nonetheless, in our anti-money laundering example, we use an existing detection system (originally based on simple rules) and therefore could experiment with an additional baseline where the generator is trained to find adversarial examples for this existing detection system. We will strongly consider this in a next revision, since this may convey the difference between the typical approach and our method more clearly to future reviewers.
> > >
> > > **Lack of theoretical analysis**
> > >
> > > We agree that there is no theoretical analysis. However, this need not be a huge drawback if we can show empirical usefulness. Moreover, we believe that conceptually our method is not that complicated, we add an extra objective to the GAN architecture that nudges the generator towards the generation of examples with certain properties. One can think of the two objectives (GAN vs malicious objective) having an opposite effect on the generator: the former wants to keep the generated data in distribution, the latter wants to push the generated data out of distribution. The hyperparameter balances the strength of these opposing goals, and we perform hyperparameter searches to identify suitable ranges. This idea is not hard to grasp and it is clear to understand why this might work, as we also argue in the article. We do agree that the method requires domain expertise. While our recipe is general, malicious activities in various domains have different properties and the objective function should therefore be carefully defined. We agree that the methods can be clarified and expanded.
> > >
> > > We would argue that significant tuning is a typical requirement in most deep learning applications, and our method barely impacts that requirement by the additional two hyperparameters.
> > >
> > > We therefore feel it is most unfair that the reviewer judges our work as unscientific. In our opinion this is not a term that should be used lightly by a reviewer, e.g. only in cases where the experimental setup is fundamentally flawed, and not because of the usage of GANs or the claims of generality, which are minor comments that can be easily addressed in revised versions.  We sincerely hope that the reviewer revises this remark. Moreover, given our responses to the previous remarks, we feel that the reviewer did not grasp the key aspect of our method.
> > >
> > > **Misleading evaluation of “real-world”**
> > >
> > > We thank the reviewer for their comment. We did not wish to fool the reader, and we used the term real-world to contrast with the common usage of synthetic data in the financial and anti-money laundering domain. We will clarify this in future versions. We can confirm that the anti-money laundering dataset is a real-world (but private) banking dataset, which is as close to a deployment setting as we can get in this proof-of-concept stage. The recommender dataset is indeed a benchmark dataset.

---

> > > > ### Author Response · Authors · 2022-11-09
> > > > **Continued clarifications**
> > > >
> > > > We thank the reviewer for pointing out various spelling mistakes and paragraph transitions, and will make sure to adapt this in the future versions. We will also expand the methods sections to improve clarity.
> > > >
> > > > **the term “adversarial” is misleading, and it is not true that “attacker and defence system constantly adapt” (I dare say in most domains there is no adaptation by the “defence” at all). I suggest revising this paragraph entirely.**
> > > >
> > > > We disagree that the term adversarial is misleading however, and in our domain (finance) we can confirm that the defense systems are continuously updated and the attackers continuously change strategies accordingly. We will clarify in future versions that this may not always be the case in other domains.
> > > >
> > > > **I checked (Lannoo & Parlour, 2021) and I couldn’t find a single occurrence of “0.7—3 trillion”**
> > > >
> > > > We thank the reviewer for the comment. The introduction of the cited report states that €1.7-4 trillion (between 2 and 5% of global GDP) are laundered annually, while only around 1.1% is detected. We may have erroneously interpreted the 1.1% as part of the global GDP, resulting in the figures mentioned in our introduction. Instead, the situation is even worse, as the 1.1% of recovered funds applies to the estimated money being laundered. We should therefore update the cited numbers to be close to €1.7-4 trillion.
> > > >
> > > > **What is the goal of generating “out of distribution” samples if such samples are not realistic in the first place?**
> > > >
> > > > We thank the reviewer for the comment. This is indeed not the intended interpretation and we agree that the way it is written causes confusion. What was intended by ‘realistic data’ is data generated from the same distribution as the available dataset. This is different in our work, since we are not looking to generate data from such a distribution (which would result in legitimate examples), but instead nudge the generator to produce examples which have properties of malicious activity. We will clarify this in future versions.

---

> > > > > ### Comment · Reviewer_hdrh · 2022-11-10
> > > > > **Ack 3**
> > > > >
> > > > > ### Audience and Recommendation
> > > > >
> > > > > As a final piece of advice, I must stress that my entire review was done from the viewpoint of a reviewer who is reviewing a paper submitted to ICLR. Personally, I would have had little against "recommending acceptance" of this paper had it been submitted to a workshop. At the same time, my criticism would have been less stark if the paper had been submitted to an engineering venue: ultimately, the interesting part of the paper is a method that (seems to) work in a (specific) setting. I am confident that some venues would greatly appreciate a similar paper. But not ICLR (at least in my opinion).
> > > > >
> > > > > I endorse the authors in pursuing this line of research. However, I invite them to tone-down the paper, and focus on the specific contribution---which must be substantiated properly. Such contribution would be easier to support if the authors focused their attention on a single domain: I believe that finance is a very attractive research field, and the if the authors are experts in this area, then they should focus their efforts on this single domain instead of aiming at generality. Furthermore, the authors should better substantiate the necessity of labels, thereby improving the real-world value of TheGanfather (for finance).
> > > > >
> > > > > ### Summary
> > > > >
> > > > > In summary, I appreciate the authors' efforts but I am still unable to recommend acceptance. I may increase my score (slightly), in light of the positive impression of the authors' responses. However, despite the clarifications, I am still convinced that the paper is not yet ready for ICLR since most of the claims are not supported.
> > > > >
> > > > > (do note that the "poor novelty value" is, by far, the least of my concerns. The problem of the paper lies not in the fact that the ```GANFather``` is (arguably) not novel -- but rather, in the fact that most of the claimed contributions are either misleading or not well supported, and the overall paper fits not ICLR)

---

> > > > ### Comment · Reviewer_hdrh · 2022-11-10
> > > > **Ack 2**
> > > >
> > > > ### The contributions are not convincing
> > > >
> > > > The primary reason for my "negative" recommendation is that the paper makes very bold claims, some of which are not contributions (and I've already explained my reasonings); while the others are (i) not supported with evidence, and (ii) not even defined in such a way to make the claims "falsifiable". The fourth claimed "contribution" is  ```our method is general```, which is grounded on a mere ```we validate our method in two domains```. Setting aside that a "validation on two domains" is far from being enough to claim that the method is indeed "general", what is even questionable is that such "validation" is purely empirical. The results show an improvement over the "baselines", but this is not enough to support such a claim. To make things worse, such claim is "not falsifiable": on what basis can one accept or refute the claim that the method is "general"? This is why I used the term "unscientific" in my review.
> > > >
> > > > On this note, I must correct the authors, and I invite them to re-read my comment, which does not ```judge(s) (y)our work as unscientific```. What is unscientific is *my interpretation* **of the paper**$^1$, which stems from what it is written **in the paper**$^2$. It is unscientific to claim that "Our method is general" (fourth contribution, taken verbatim from **the paper**) and then not supporting such contribution with any proof -- aside from an evaluation of questionable value (as also remarked by the other reviewers) conducted on just two datasets. Supporting such a strong claim requires either a detailed and well-principled theoretical proof/argument, or an extremely large set of experiments. The paper does not provide either --- meaning that it is even impossible to "prove that such a claim is wrong" (because it is unfalsifiable), and hence "unscientific".
> > > >
> > > > In truth, I endorse the authors to tone-down the "generality" and tailor their proposal on a single domain (e.g., finance?), and then perform various (but thorough) experiments on systems (and data) of this single domain.
> > > >
> > > > $^1$ Actually, my "unscientific" is specifically addressed at (my interpretation of) the "claimed generality" -- and not even of the paper as a whole.
> > > >
> > > > $^2$ Put differently, it is up to the authors to convince me -- the reviewer (and the reader) -- that the paper is a valuable scientific contribution. My impression after reading **the paper** is that this is not the case.

---

> > ### Comment · Reviewer_hdrh · 2022-11-10
> > **Ack**
> >
> > I thank the authors for their response, and I acknowledge their clarifications. In particular, I am pleased to notice that such clarifications have been written with great care -- surprisingly, superior to the one given to the actual paper. While my opinion on the paper is still "negative", I appreciate the authors' efforts in responding, and this leads me to believe that their contribution will (eventually) become a worthy addition to the state-of-the-art.
> >
> > In what follows, I will respond to some of the comments made by the authors. My ultimate goal is making this paper into a valuable contribution to the state-of-the-art (be it for ICLR, or for any future "developments").
> > Before I do this, however, I must anticipate that my review was (heavily) criticizing **the paper** -- and not *the authors*. Indeed, all my remarks were due to the paper being not convincing enough in any of its major claims. On the contrary, I am confident (and even more so after the author's responses) that the authors spent a lot of effort. It is a pity that **the paper** does not gave enough credit to the authors' endeavours. Yet, as a reviewer, I can only evaluate what I am given -- which is **the paper**. (I am emphasizing this because I want to avoid any potential misunderstandings).
> >
> > ### Labels
> > I acknowledge the authors' response, but my critique still stands. In my review, I stated that it was misleading to claim the lack of labels as "a contribution". As also remarked by the other reviewers, the proposed solution simply entails leveraging a well-known technique (i.e., a GAN -- which by default does not require labels) to generate "out of distribution" adversarial examples. The lack of labels is clearly "good", but it is not "a contribution" -- but "a property" of the proposed solution (which is based on prior art). Therefore, it is still unfair to claim the lack of labels as "a contribution".
> >
> > Also, I have remarks on this sentence ```this requirement is clearly of a different scale than labeling thousands or millions of potentially suspicious activities```: is there hard evidence showing that it is necessary to label "thousands or millions" of potentially suspicious activities? Note that by "evidence" I do not mean "a paper that performs some experiments on millions of samples", but rather "a paper proving that thousands/millions of samples are *needed*" (also, the paper must be on the same domain).
> >
> > ### Adversarial Attacks
> > The authors' state that ```the reviewer erroneously interpreted the purpose of our work to be a typical ‘adversarial attack’ setting where the goal is to find the smallest possible perturbations to known samples that lie on the other side of the decision boundary of a trained classifier.```. I assure them that this is not the case: I was perfectly aware that the perturbations may very well not be "smallest" (as a matter of fact, the perturbations entailed in some of the works I referenced in my review are very large). Nonetheless, the very first contribution of the paper states ```We propose a framework to generate synthetic data, simulating adversarial attacks by malicious actors```. If the intention was to consider different types of attacks w.r.t. "traditional adversarial attacks", then I invite the authors to clearly specify their intentions.
> >
> > Furthermore, I have remarks about ```in our domain (finance) we can confirm that the defense systems are continuously updated```. What does "continuously" mean? Does it mean that a system is updated whenever a new attack is proposed? Applying patches when they are released can hardly be considered as ```constantly adapt to the (attacker's) behavior```. Talking about finance: a few months ago a paper showed the weakness of ML-based algorithmic trading to adversarial perturbations (Nehemya, Elior, et al. "Taking Over the Stock Market: Adversarial Perturbations Against Algorithmic Traders." ECML-KDD, 2021.). Are real systems patched already against these attacks?
> >
> > Put simply, I believe that the first sentence of the paper is an (unnecessary) overstatement, which requires some supporting evidence.

---

### Official Review · Reviewer_kgFf · 2022-10-25

**Confidence:** 4
**Clarity, Quality, Novelty And Reproducibility:** original work
**Correctness:** 2
**Technical Novelty And Significance:** 1
**Empirical Novelty And Significance:** Not applicable
**Recommendation:** 1

**Strength And Weaknesses:**

Strengths:

1. Compared with classical GANs,the loss of the author’s generator  is a linear combination of the loss of three components,including the optimisation objective,the GAN and the alert system .this makes the author’s framework can produce out-of-sample data without label requirements.

2. In two real-world  cases, money laundering, and recommendation systems.the generator of framework can produce meaningful attacks ,the discriminator can obtain near-perfect classification by training.


Weaknesses:

1. The method is  not well demonstrated either by theory or practice.For example, in section 2.1.1,the paper points that the loss of generator of the framework is a linear combination ,where the weights are three hyperparameters controlling. It lacks  theoretical proof in this paper.

2. This method has poor versatility because the structure of generator and the optimisation objective of the framework need to be customized.

3. In the experiments,the discriminator can obtain near-perfect classification,but the author do not provide convincing evidence of the correctness of the proposed approach or its utility compared to existing approaches.


**Summary Of The Paper:**

This paper introduces an adversarial method without label requirements, to enhance the detection of illicit activity in various domains.This method comprises a generator that produces meaningful attacks and a discriminator to detect them.This meaningful attacks can reveal  defensive weaknesses for the discriminator to correct.The method  is evaluated with a suite of experiments in money laundering and recommendation systems .

**Summary Of The Review:**

The author provides very limited theoretical and empirical results.This submission looks more like a working draft rather than a conference paper.

---

> ### Author Response · Authors · 2022-11-09
> **Clarifications on the claims**
>
> Please find our replies to the above concerns:
>
> 1. We thank the reviewer for the feedback. We agree there is no theoretical proof of the choices, rather we justify adding the malicious objective to nudge the generator to generate data with some desired properties. One can think of the two objectives (GAN vs malicious objective) having an opposite effect on the generator: the former wants to keep the generated data in distribution, the latter wants to push the generated data out of distribution. The hyperparameter balances the strength of these opposing goals, and we perform hyperparameter searches to identify suitable ranges. We do not agree that a lack of rigorous theoretical justification is a strong weakness if the intuitive justification is sound and the empirical results are interesting. Nonetheless, we will try to add more rigorous justification and theoretical analysis in future versions.
>
>
> 2. We agree with the reviewer that our method needs to be customized depending on the use-case at hand. This is indeed a weakness that is hard to overcome, since each use-case may have its own characteristics. However, one could argue that a typical GAN setup also requires customization depending on whether it is used on images, videos, sounds, language etc. We therefore do not see this as a major limitation, and believe that our recipe of adding the malicious loss and an optionally existing detection system to a GAN architecture is generally applicable.
>
> 3. Regarding the first point, we agree that it would be great to test our setup in a deployment setting and have feedback from human analysts. While we are planning to do this in the future, we believe that the proof-of-concept provided in our work contains enough value by itself.
> Regarding the second point, we believe that the difference between our setup and a typical GAN setup is the main utility. Adversarial attacks in existing approaches typically try to find the smallest possible perturbations to known samples that lie on the other side of the decision boundary of a trained classifier. Importantly, in this approach labels are required to train the classifier in the first place. We avoid the label requirement by instead stimulating the generator to produce OOD examples that optimize a new objective. In this setup, we both guarantee that no labels are needed and that the generated samples have properties of malicious activity, which are two main differences from current approaches.

---

### Official Review · Reviewer_x8ad · 2022-10-26

**Confidence:** 4
**Correctness:** 3
**Technical Novelty And Significance:** 2
**Empirical Novelty And Significance:** 2
**Recommendation:** 3

**Clarity, Quality, Novelty And Reproducibility:**

The paper is written clearly and I have not checked the code for reproducibility, but the authors present enough details.

I believe the novelty is limited - see weaknesses listed.


**Strength And Weaknesses:**

The idea here is not novel as there are a number of papers that use a GAN like structure for generating OOD samples (or adversarial examples) - see references below, of course, mostly focussing on image domain. This paper does not work on image domain, which is actually a strength of the paper.

But, I did not see any insights that are generalizable or even some domain specific hard structural problem that is addressed (i.e., the loss functions used are quite natural and probably too simple). I am not sure why these objective were chosen, is there some citation to support these? I was also hoping to see some quantitative measure of how good the attack is, which in the paper is just shown qualitatively

I do not understand the part about discriminator - a line in the paper says that "the discriminator eventually learns to distinguish synthetic attacks from real data", but isnt a GAN supposed to eventually make the discriminator not be able to distinguish (by learning a good generator)? Along same lines, I do not understand what is going on in Figure 4? It seems rather like adversarial training that a neural network is trained with OOD data and is able to then detect those. But, then the generator can be trained again to bypass this new discriminator?

Baluja, S., & Fischer, I. (2018). Learning to Attack: Adversarial Transformation Networks. Proceedings of the AAAI Conference on Artificial Intelligence, 32(1).
Liu, A., Liu, X., Fan, J., Ma, Y., Zhang, A., Xie, H., & Tao, D. (2019). Perceptual-Sensitive GAN for Generating Adversarial Patches. Proceedings of the AAAI Conference on Artificial Intelligence, 33(01)
Xiao, C., Li, B., Zhu, J. Y., He, W., Liu, M., & Song, D. (2018). Generating adversarial examples with adversarial networks. In 27th International Joint Conference on Artificial Intelligence, IJCAI 2018 (pp. 3905-3911). International Joint Conferences on Artificial Intelligence.

**Summary Of The Paper:**

The authors propose a GAN based approach to generate attacks and also claim to enhance the detection of illicit activity in various domains. They use three type of loss functions to train a generator to generate the attacks. The domain they focus on is money laundering and recommendation systems. The authors claim there methods applies to other settings

**Summary Of The Review:**

My recommendation is based on my perception of limited novelty and lack of new generalizable principles to be taken away from this work.

---

> ### Author Response · Authors · 2022-11-09
> **Differences to usual 'adversarial attack' approaches**
>
> Please find our replies to the above concerns:
>
> **On the novelty:**
>
> We thank the author for the comments. We acknowledge that there is a large literature regarding the generation of adversarial examples. However, we would like to emphasize the difference between the ‘adversarial attack’ literature and our proposed work:
>
> Adversarial in the former context regards finding the smallest possible perturbations to known samples across the decision boundary of a trained classifier. (e.g. for money laundering, this would translate to finding the smallest perturbation of a money laundering sample such that the classifier would label it as legitimate). Importantly, in this approach labels are required to train the classifier in the first place.
>
> In domains such as money laundering, labeled datasets are virtually non-existent. Our approach is specifically designed to avoid the need for any labels. The way this is achieved, is by adding an extra objective for the generator, characterizing the malicious intent. In this way, starting from an unlabeled dataset assumed to contain predominantly legitimate samples, we still manage to generate samples with properties of malicious activity.
>
> **On the generalizability, loss functions and quantitative analysis:**
>
> We thank the reviewer for the feedback. As explained in our previous response, the main difference between related work and our method is the label requirement. We will make this difference more clear in future versions, since this is an essential property to understand when valuing our work.
>
> The objective functions are as natural and simple as possible for characterizing the malicious activity. We specifically chose to keep them as simple as possible at first, but there are no constraints to add more complexity as long as they remain differentiable. We will add more discussion around the choice of objective functions.
>
> Regarding the quantitative measures: in the anti-money laundering case, we show in Figure 10 how much money is flowing through the accounts undetected in the generated vs the legitimate samples. We show how our generated samples are much better at moving money through the accounts, while still not triggering the existing defense system. In the recommender system case, we show in Figure 5 how much our generated attacks lead to top-10 recommendation compared to naive baselines. In both cases we have therefore quantitatively studied how well the malicious objective is fulfilled. We will try to make this more explicit in future versions and expand the analysis.
>
> **On the distcriminator:**
>
> We thank the reviewer for this important point. The first question points to a fundamental difference between our setup and a typical GAN setup. In our setup, the additional objective added to the generator competes with the gradients coming from the discriminator. The former tries to push the generated data away from the real data distribution, while the latter tries to align the generated data distribution with the real data distribution. Importantly, our concern is not to generate data that exactly follows the data distribution. Hence, we are not requiring the discriminator to converge to random accuracy.
>
> The goal of our method differs from typical GANs, where we try to generate OOD data that is not too different from real data. While we did not perform a theoretical analysis of this setup, we empirically performed hyperparameter optimization to find regions of where the generated data maximizes the malicious objective while not triggering  the existing defense system. We found that the discriminator typically maintains near-perfect classification performance, which in turn is exactly what we are after because the discriminator can now be used to improve the existing defense. In any case, we will try to add more theoretical justification of our approach.
>
> Regarding Figure 4, the reasoning is more subtle. A generator may suffer from mode collapse (a single “winning” strategy), the discriminator paired with a certain generator may specialize in only detecting that strategy. To test this, we built a dataset using synthetic samples from various generators (trained with various hyperparameters and seeds). We then tested the classification performance of the (already trained) discriminators on this mixed dataset. A large number of discriminators still achieve near-perfect performance, indicating detection of a variety of attacks. No generator or discriminator is retrained in this scenario. In the recommendation system, the results were different: the trained discriminators achieved around 0.75 AUC on the mixed dataset. In this scenario, mode collapse may have been more prominent. We showed that one can train a new discriminator on the mixed dataset and achieve near-perfect performance. It would be interesting to assess why mode collapse is a larger problem in one scenario. We will make these arguments more explicit in future versions.

---

### Author Response · Authors · 2022-11-18
**New version**

Dear reviewers,

We thank you again for all your feedback and suggestions. We have uploaded a revised version of our manuscript, with the following main changes:

- Addition of Figure 1, to contrast our approach with previous work.
- Addition of theoretical justification of our method in Appendix A3.
- Toning down / clarifying the claims.
- More detailed Methods and Conclusions.
- Polishing text throughout.

We hope that these changes address as much as possible your previously stated concerns. We will be glad to hear your feedback.

---

> ### Author Response · Authors · 2022-12-12
> **New version (2)**
>
> Dear reviewers,
>
> We thank reviewer 3 to reconsidering the recommendation score to 6 after revising our changes. We hope that other reviewers also appreciate the new draft and in any case we are happy to receive feedback.
>
> Kind regards,

---

### Decision · Program_Chairs · 2023-01-20

**Decision:**

Reject

**Justification For Why Not Higher Score:**

Limited novelty/generalizability of results

**Justification For Why Not Lower Score:**

N/A

**Metareview: Summary, Strengths And Weaknesses:**

To tackle the problem of missing or delayed labels in adversarial settings, the authors proposed The GANfather, an adversarial and label-free method to both (1) generate a variety of meaningful attacks, as guided by a custom, user-defined objective function; and (2) train a defence system to detect such attacks. The authors test their framework in two real-world use-cases, namely injection attacks in recommendation systems and anti-money laundering.

Though reviewers liked the setting, they found the paper to be a bit too applied in nature and limited in scope: it is not clear how well it would apply to other domains, datasets, etc., or whether there is a deeper/more fundamental conclusion to be made from this (fundamentally experimental) paper. They authors made changes and toned down their claims, but reviewer concerns remained even after the rebuttal.

**Summary Of Ac-Reviewer Meeting:**

N/A